Aberrant repair initiated by the adenine-DNA glycosylase does not play a role in UV-induced mutagenesis in Escherichia coli

Zutterling Caroline 1
Mursalimov Aibek 2
Talhaoui Ibtissam 3
Koshenov Zhanat 2
Akishev Zhiger 4
Bissenbaev Amangeldy K. 4
Mazon Gerard 3
Geacintov Nicolas E. 5
Gasparutto Didier 6
Groisman Regina 1
Zharkov Dmitry O. 7 8
Matkarimov Bakhyt T. 2
Saparbaev Murat 1 murat.saparbaev@gustaveroussy.fr
1 Groupe «Réparation de l’ADN», Equipe Labellisée par la Ligue Nationale Contre le Cancer, CNRS UMR8200, Université Paris-Sud, Gustave Roussy Cancer Campus , Villejuif , France
2 National Laboratory Astana, Nazarbayev University , Astana , Kazakhstan
3 CNRS UMR 8200—Laboratoire «Stabilité Génétique et Oncogenèse», Université Paris Sud (Paris XI), Gustave Roussy Cancer Campus , Villejuif , France
4 Department of Molecular Biology and Genetics, al-Farabi Kazakh National University, Faculty of Biology , Almaty , Kazakhstan
5 Chemistry Department, New York University , New York City, NY , USA
6 CEA, CNRS, INAC, SyMMES, Université Grenoble Alpes , Grenoble , France
7 SB RAS Institute of Chemical Biology and Fundamental Medicine , Novosibirsk , Russia
8 Novosibirsk State University , Novosibirsk , Russia
Vassetzky Yegor
Electronic publication date: 2018 Dec 5
Publication date: 2018
Volume: 6
Electronic Location ID: e6029
Received 2018 Sep 12; Accepted 2018 Oct 30
Copyright: © 2018 Zutterling et al.
Copyright year: 2018
Copyright holder: Zutterling et al.
License: This is an open access article distributed under the terms of the Creative Commons Attribution License, which permits unrestricted use, distribution, reproduction and adaptation in any medium and for any purpose provided that it is properly attributed. For attribution, the original author(s), title, publication source (PeerJ) and either DOI or URL of the article must be cited.
License URL: https://creativecommons.org/licenses/by/4.0/

Keywords: Adenine-DNA glycosylase, Nucleotide excision repair, UV-induced mutagenesis, Cyclobutane pyrimidine dimer, Base excision repair, Aberrant DNA repair, Escherichia coli, Pyrimidine (6–4) pyrimidone photoproduct

Funding: Murat Saparbaev from la Ligue National Contre le Cancer “Equipe Labellisee,” Electricité de France RB 2017 French National Center for Scientific Research PRC CNRS/RFBR n1074 REDOBER NU ORAU Science Committee of the Ministry of Education and Science of the Republic of Kazakhstan Program 0115RK03029 Russian Ministry of Science and Education 6.5773.2017/6.7 Russian Science Foundation 17-14-01190 Science Committee of the Ministry of Education and Science of the Republic of Kazakhstan AP05131598 Arcane Labex program, funded by the French National Research Agency ARCANE project no. ANR-12-LABX-003 Postdoctoral fellowships from the Fondation ARC This work was supported by grants to Murat Saparbaev from la Ligue National Contre le Cancer “Equipe Labellisee,” Electricité de France (RB 2017) and French National Center for Scientific Research (PRC CNRS/RFBR n1074 REDOBER); and to Bakhyt T. Matkarimov from NU ORAU (https://nu.edu.kz/) and Science Committee of the Ministry of Education and Science of the Republic of Kazakhstan, Program 0115RK03029; and to Dmitry O. Zharkov from the Russian Ministry of Science and Education (6.5773.2017/6.7) and Russian Science Foundation (17-14-01190), and to Amangeldy K. Bissenbaev from the Science Committee of the Ministry of Education and Science of the Republic of Kazakhstan [grant No. AP05131598], and to Nicolas E. Geacintov from US NIEHS Grant ES024050. Didier Gasparutto received support from the Arcane Labex program, funded by the French National Research Agency (ARCANE project no. ANR-12-LABX-003). Ibtissam Talhaoui was supported by postdoctoral fellowships from the Fondation ARC. The funders had no role in study design, data collection and analysis, decision to publish, or preparation of the manuscript.

==============================
Background

DNA repair is essential to counteract damage to DNA induced by endo- and exogenous factors, to maintain genome stability. However, challenges to the faithful discrimination between damaged and non-damaged DNA strands do exist, such as mismatched pairs between two regular bases resulting from spontaneous deamination of 5-methylcytosine or DNA polymerase errors during replication. To counteract these mutagenic threats to genome stability, cells evolved the mismatch-specific DNA glycosylases that can recognize and remove regular DNA bases in the mismatched DNA duplexes. The Escherichia coli adenine-DNA glycosylase (MutY/MicA) protects cells against oxidative stress-induced mutagenesis by removing adenine which is mispaired with 7,8-dihydro-8-oxoguanine (8oxoG) in the base excision repair pathway. However, MutY does not discriminate between template and newly synthesized DNA strands. Therefore the ability to remove A from 8oxoG•A mispair, which is generated via misincorporation of an 8-oxo-2′-deoxyguanosine-5′-triphosphate precursor during DNA replication and in which A is the template base, can induce A•T→C•G transversions. Furthermore, it has been demonstrated that human MUTYH, homologous to the bacterial MutY, might be involved in the aberrant processing of ultraviolet (UV) induced DNA damage.

Methods

Here, we investigated the role of MutY in UV-induced mutagenesis in E. coli. MutY was probed on DNA duplexes containing cyclobutane pyrimidine dimers (CPD) and pyrimidine (6–4) pyrimidone photoproduct (6–4PP). UV irradiation of E. coli induces Save Our Souls (SOS) response characterized by increased production of DNA repair enzymes and mutagenesis. To study the role of MutY in vivo, the mutation frequencies to rifampicin-resistant (RifR) after UV irradiation of wild type and mutant E. coli strains were measured.

Results

We demonstrated that MutY does not excise Adenine when it is paired with CPD and 6–4PP adducts in duplex DNA. At the same time, MutY excises Adenine in A•G and A•8oxoG mispairs. Interestingly, E. coli mutY strains, which have elevated spontaneous mutation rate, exhibited low mutational induction after UV exposure as compared to MutY-proficient strains. However, sequence analysis of RifR mutants revealed that the frequencies of C→T transitions dramatically increased after UV irradiation in both MutY-proficient and -deficient E. coli strains.

Discussion

These findings indicate that the bacterial MutY is not involved in the aberrant DNA repair of UV-induced DNA damage.

Introduction

In all cellular organisms DNA composition is limited to only four nucleotides C, G, A and T, except for DNA methylation, which is appeared as a protection mechanism against foreign DNA or as a gene regulatory system. It is noteworthy that this strict evolutionary constraint also enables DNA repair systems to distinguish between damaged and normal bases. DNA glycosylases recognize and excise modified bases among vast majority of normal bases via the base excision repair (BER) pathway. Nevertheless, a mispair composed of two normal bases occuring due to spontaneous deamination of 5-methylcytosine to thymine and DNA polymerase errors during replication presents a challenging puzzle to repair systems.

Hydroxyl radicals (OH•) preferentially react with C8 atom of purines to generate 7,8-dihydro-8-oxoguanine (8oxoG) in DNA and 8-oxo-2′-deoxyguanosine-5′-triphosphate (8oxodGTP) in nucleotide pool. 8oxoG is a highly mutagenic DNA adduct considered as a major oxidative DNA lesion in aerobic organisms. To counteract the genotoxic effects of oxidized guanines, organisms have evolved multiple overlapping DNA repair and error-avoiding mechanisms. The majority of oxidative DNA base lesions including 8oxoG are removed in the BER pathway (Barnes & Lindahl, 2004). BER is initiated by a DNA glycosylase which cleaves the N-glycosydic bond between the abnormal base and the deoxyribose, generating either an abasic site or single-stranded break in DNA. In E. coli, the system referred as GO pathway, which consist of three enzymes: the bi-functional 8oxoG-DNA glycosylase/AP lyase Fpg/MutM which excises the 8oxoG opposite cytosine (Tchou et al., 1991), MutY which excises adenine misincorporated across unrepaired 8oxoG in DNA template (Michaels et al., 1992) and finally MutT which cleanses cellular dNTPs pool by hydrolyzing 8oxodGTP precursors, and thus avoiding its incorporation into DNA (Maki & Sekiguchi, 1992). The crystal structure and mutagenesis studies revealed that MutY cleaves the N-glycosydic bond through a hydrolytic mechanism requiring Asp138, followed by an inefficient β-elimination reaction which is structurally and chemically uncoupled from the initial glycosydic bond cleavage (Guan et al., 1998; Manuel et al., 2004). Nevertheless, consistent with the established mechanism of action of bi-functional DNA glycosylases/AP lyases, MutY generates a Schiff base intermediate between the nucleophilic lysine and C1′ of the AP site (Sun et al., 1995). Multiple evidence demonstrates that the GO pathway prevents mutations induced by oxidation of guanines in DNA and nucleotide pool in many different species. For example, the E. coli fpg mutY double mutant exhibits an extreme mutator phenotype with 10,000-fold increase in G•C→T•A transversions that can be reversed by plasmids carrying the Fpg gene (Duwat et al., 1995). At the same time, E. coli mutT also shows a strong mutator phenotype, characterized by the specific increase in A•T→C•G transversions (Yanofsky, Cox & Horn, 1966). Intriguingly, the presence of mutY mutation lowers the frequency of A•T→C•G spontaneous mutations in both E. coli wild-type (WT) and mutT strains, suggesting that MutY can act in mutagenic post-replication repair pathway by excising correct A in the template DNA strand when it is opposite a misincorporated 8oxodGTP residue (Fowler et al., 2003).

MYH (MUTYH) is a human homologue of the E. coli MutY protein, which displays very similar DNA substrate specificities to the bacterial counterpart. Mutations in MUTYH gene are associated with certain types of familial colorectal tumors without a germ-line mutation in the APC gene and they also confer a spontaneous mutator phenotype in human and mice cell lines (Al-Tassan et al., 2002). Previously, Vrouwe et al., (2011) have shown that exposure of the non-cycling NER-deficient XP-C and XP-A human fibroblasts to ultraviolet (UV) radiation resulted in the generation and accumulation of single-strand DNA breaks 24 h after treatment, which in turn activated ATR-dependent DNA damage response. Intriguingly, the formation of single-strand DNA breaks at damage sites and DNA repair synthesis initiated by these breaks did not lead to removal of UV adducts in XP fibroblasts. However, recently Mazouzi et al., (2017) revealed that the presence of MUTYH protein in NER-deficient cells results in an increased sensitivity to UV light. The authors proposed that MUTYH may inhibit the alternative NER-independent repair of UV-induced DNA lesions in XP cells. Nevertheless, it might be also possible that human adenine-DNA glycosylase targets A opposite UV adducts and cause aberrant futile repair of the non-damaged DNA strand. Thus, one could imagine that the severe phenotype of XP patients might be due to a DNA glycosylase-initiated aberrant repair of UV-induced DNA lesions in human cells that would lead to an extremely mutagenic scenario.

Ultraviolet radiation generates two most common DNA lesions: the cyclobutane pyrimidine dimer (CPD) and the pyrimidine (6–4) pyrimidone photoproduct [(6–4)PhotoProduct; 6–4PP]. Both photoproducts are cytotoxic (block DNA replication and transcription) and mutagenic, while CPDs are several times more frequent than 6–4PPs (Douki et al., 2000). A hallmark of UV mutagenesis is the high frequency of C→T transitions at dipyrimidine sites in DNA, possibly due to the extremely high deamination rate of cytosine residues within CPD sites in DNA (Peng & Shaw, 1996). UV radiation greatly induces mutagenesis in E. coli, however mutation in the lexA, recA and umuD(C) abolish UV mutagenesis implying the existence of error-prone DNA repair mechanisms. Following the pioneering studies of Witkin and Radman’s hypothesis on SOS repair, work by the numerous laboratories have established and characterized to a great detail the E. coli SOS response system. According to the widely accepted model, following DNA damage, RecA becomes activated in the presence of single-stranded DNA regions and mediates LexA proteolytic cleavage which in turn leads to the increased expression of more than 40 genes including recA, uvrA and umuDC. Genetic and biochemical evidence suggests that SOS mutagenesis is largely the result of the action of error-prone translesion DNA polymerases, such as Pol V (encoded by the umuDC operon), which have the ability to insert nucleotides opposite various DNA lesions and thus enabling the lesion bypass and restart of stalled DNA replication forks by replicative DNA polymerases (Patel et al., 2010).

In our previous studies, we have established the existence of the aberrant repair pathway, initiated by the human mismatch-specific thymine-DNA glycosylase (TDG), which can target the non-damaged DNA strand and excise thymine when it is paired with a damaged adenine residue in DNA duplex (Talhaoui et al., 2014). In vitro reconstitution of BER with duplex DNA containing hypoxanthine opposite T (Hx•T pair) and TDG resulted in the incorporation of cytosine opposite Hx. Based on the mutagenic properties of mismatch-specific DNA glycosylases we proposed that the aberrant BER pathway initiated by MutY in E. coli and TDG and MUTYH in mammalian cells toward unrepaired DNA lesions may lead to the mutation fixation in absence of DNA replication (Talhaoui et al., 2017). Here, we characterized in vitro the DNA glycosylase activity of E. coli MutY protein toward short oligonucleotide duplexes containing UV-DNA adducts. In addition, we analyzed mutation rates and spectra in E. coli strains lacking MutY after exposure to short-wavelength UV radiation. The role of MutY-catalyzed DNA glycosylase activity in the UV-induced mutagenesis is discussed.

Methods Summary

Bacterial strains, plasmids and enzymes

All bacterial genetics procedures were performed as described (Miller, 1972); these included preparation of media and mutagenesis. The rich medium was the LB broth, supplemented when required with either 100 μg·μL−1 ampicillin, or 50 μg·μL−1 kanamycine, or 100 μg·μL−1 rifampicin. E. coli cells were transformed by DNA electroporation as described (Dower, Miller & Ragsdale, 1988).

The E. coli K12 strains used in this study are listed in Table 1. Original Miller’s strain CC104, generously provided by Dr J.H. Miller (University of California, USA), is derivative of P90C[ara, Δ(lac proB)XIII] carrying a lacZ allele on an F′(lacI–, Z– proB+) plasmid and, therefore, exhibiting the Lac– phenotype (Cupples & Miller, 1989). Strains AB1157 and CC104 (wild type for DNA repair), BH200 carrying mutation in the UvrA gene, BH980 in the MicA (also referred as MutY) gene, BH1070 and BH1220 containing double mutation uvrA6 micA were from laboratory stock. Strain AK146 carrying mutation in the UvrA gene was generously provided by Dr. A. Kuzminov (University of Illinois, Champaign, IL, USA). Strains were grown overnight at 37 °C 200 rpm. Cultures were then plated (stripped) on LB-agar plates and incubated overnight to get single colonies.

Table 1 Bacterial strains.

Strain	Genotype	Source of derivation	
AB1157	thr-1 leu-6 proA2 his-4 argE3 thi-1 lacYI galK2 ara-14 xyl-5 mtl-1 tsx-33 rpsL31 supE44	Laboratory stock	
BH200	as AB1157 but uvrA::Tn10	Laboratory stock	
AK146	as AB1157 but uvrA6	A. Kuzminov (University of Illinois, USA).	
BH1220	as AB1157 but uvrA6 micA::KnR	Laboratory stock	
BH1070	as GC4468 F− Δlac4169 rpsL but uvrA6 micA::KnR	Laboratory stock	
CC104	P90C [araΔ(lacproB)XIII] carrying an F′lacI-Z–proB+ episome (G•C→T•A)	J.H. Miller (University of California, USA).	
BH980	as CC104 but micA::KnR	Laboratory stock	

The plasmid pJWT21-4 encoding the E. coli MutY/MicA protein was generously provided by Dr. P. Radicella (CEA, Fontenay-aux-Roses, France). Site-directed mutation aspartic acid 138 → asparagine (D138N), within the MutY coding sequence in pJWT21-4 were generated by the QuikChange site-directed mutagenesis kit according to the manufacturer’s instructions (Stratagene Europe, Amsterdam, Netherlands). Following oligonucleotide primers were used to generate MutY-D138N mutant: forward primer, d(CACTTTCCGATTCTCAACGGTAACGTCAAACG) and reverse primer, d(CGTTTGACGTTACCGTTGAGAATCGGAAAGTG).

UV-damage endonuclease (UVDE) from Saccharomyces pombe and T4 endonuclease V (T4-PDG) were purchased from Trevigen (Gaithersburg, MD, USA) and New England Biolabs France SAS (Evry), respectively. The AP endonucleases from human (APE1) (Gelin et al., 2010) and Mycobacterium tuberculosis (MtbXthA) (Abeldenov et al., 2015) and uracil-DNA glycosylase from E. coli (UNG) (Scaramozzino et al., 2003) are from laboratory stock. The E. coli MutY protein was purified from E. coli B834(DE3) strain harbouring pET13a-MutY plasmid as described (Zharkov et al., 2000).

Preparation of plasmid DNA substrate. 70 μL of pBlueScript SK(+) plasmid DNA (0.1 μg·mL−1) in TE buffer was placed in 0.2 mL quartz cuvette and irradiated with 254 nm germicidal UV at a dose of 1,000 J·m−2 at 4 °C.

Oligonucleotides and repair assay

Oligonucleotides used in the present study are presented in the 5′→3′ direction. The 24 mer oligonucleotides containing the CPD or 6–4PP adduct: d(CTTCTTCGCAAGXXGGAGCTCTCT) where XX is either CPD or 6–4PP, and 30 mer oligonucleotide containing the CPD adduct d(AGGTCTCTTCTTCTXXGCACTTCTTCCTCC) where XX is CPD, were synthesized as described previously (Smith & Taylor, 1993). All the oligodeoxyribonucleotides containing regular DNA bases residues including their complementary oligonucleotides and also oligonucleotides containing base modifications such as Uracil (U) and 8oxoG were purchased from Eurogentec (Liège, Belgium) including the following: complementary 24 mer d(AGAGAGCTCCNNCTTGCGAAGAAG) and 30 mer d(GGAGGAAGAAGTGCNNAGAAGAAGAGACCT) where NN is either AA, AG, GA and GG dinucleotide opposite UV lesion; and modified 30 mer d(AGGTCTCTTCTTCTYYGCACTTCTTCCTCC) and 24 mer d(CTTCTTCGCAAGYYGGAGCTCTCT) or d(AGAGAGCTCCYYCTTGCGAAGAAG) and where Y is either T, U, G or 8oxoG.

The oligonucleotides were 5′-end labelled with [γ-32P]-ATP (PerkinElmer, Villebon-sur-Yvette, France) and then annealed with the complementary strands as described (Gelin et al., 2010). The standard reaction mixture (20 μL) for DNA repair assays contained five nM of [32P]-labelled duplex oligonucleotide or one μg of plasmid DNA, 20 mM Tris–HCl (pH 8.0), 100 mM NaCl, one mM EDTA, one mM DTT, 100 μg·mL−1 BSA and 100 nM MutY for 30 min at 37 °C, unless otherwise stated. The AP sites left after excision of damaged bases in synthetic oligonucleotide substrates were cleaved by incubation of the reaction mixture with light piperidine (10% (v/v) piperidine at 37 °C for 30 min and then neutralized by 0.1M HCl. Otherwise the plasmid DNA was incubated with 100 nM APE1 in DNA glycosylase buffer for 30 min at 37 °C. The samples were desalted using Sephadex G25 column (Amersham Biosciences, Little Chalfont, UK) equilibrated in 7.5M urea and the cleavage products were separated by electrophoresis in denaturing 20% (w/v) polyacrylamide gels (7 M Urea, 0.5 × TBE, 42 °C). A Fuji Phosphor Screen was exposed to the gels and then scanned with laser scanner Typhoon FLA 9500 (GE Healthcare Europe GmbH, Velizy-Villacoublay, France) and resulting digital images were analyzed using Image Gauge V3.12 software. Reaction with pBlueScript SK(+) plasmid was stopped by adding five μL of 0.25% bromophenol blue, 50% glycerol and 10 mM EDTA and the products were analyzed by 0.8% agarose gel electrophoresis (0.5X TBE).

UV mutagenesis

Strains were grown overnight at 37 °C 200 rpm and then streaked on LB-agar plates and incubated overnight (for 16 h) to obtain single colonies. Single colonies for each strain were grown in 10 mL of LB overnight at 37 °C and 200 rpm then culture was centrifuged and washed with Phosphate-buffered saline (PBS). Cells were collected by centrifugation and pellets were resuspended in the same volume of PBS and were exposed to standard germicidal UV lamp emitting at 254 nm. For this, two mL of cell suspension were irradiated in tissue culture dish with a different dose of UV irradiation at 254 nm. In general, to obtain mutation induction UV light was used at a dose that produced less than 10% survival. All procedures were performed in the dark to prevent photo-reversal of UV lesions and aliquots were taken to plate on LB plates to measure the survival. After irradiation, one mL of culture was diluted in two mL of fresh LB and incubated overnight at 37 °C 200 rpm. In the following day 0.1 mL of diluted and non-diluted overnight culture were plated on LB-agar plates containing or not 100 μg·mL−1 rifampicin which were incubated overnight at 37 °C. After incubation survived colonies were counted and data were analyzed.

Sequencing of mutations in the E. coli rpoB gene from RifR clones

The E. coli chromosomal DNA from rifampicin-resistant (RifR) clones was isolated as described (Dale & Greenaway, 1984). The DNA sequencing of the rpoB gene was performed as described (Garibyan et al., 2003). Briefly, the DNA fragment containing the main cluster II of the rpoB was obtained by PCR amplification using genomic DNA with the following oligonucleotide primers: d(CGTCGTATCCGTTCCGTTGG) and d(TTCACCCGGATAACATCTCGTC). The PCR DNA fragment was purified using the QIAquick PCR Purification Kit Qiagen kit. The purified PCR products were then sequenced with following primer d(CGTGTAGAGCGTGCGGTGAAA) using GATC service (Eurofins Genomics Company, Ebersberg, Germany). Absolute majority of the mutations (>95%) resulting in RifR clones in the present study were localized in cluster II.

Results

Construction and characterization of oligonucleotide duplexes containing UV-induced DNA adducts

To examine DNA substrate specificity of mismatch-specific DNA glycosylases we have constructed short 24 and 30 mer duplex DNA oligonucleotides containing single CPD or 6–4PP adducts at positions 13–14 in 24 mer and positions 15–16 in 30 mer. For this, 5′-[32P]-labelled modified oligonucleotides were hybridized to regular complementary strands containing two Adenine residues opposite the thymine dimers: CPD and 6–4PP. The resulting duplexes containing CPD and 6–4PP adducts were referred as AA•T=T and AA•T-T duplexes, where T=T denotes CPD and T-T denotes 6–4PP, respectively. The presence of UV-DNA adducts in the duplex DNA oligonucleotides was confirmed by their incubation in the presence of T4-PDG, which is bi-functional DNA glycosylase/AP lyase that specifically cleaves DNA duplexes containing CPD, but not that of 6–4PP adducts, and Schizosaccharomyces pombe UVDE, which recognizes both CPD and 6–4PP in duplex DNA and cleaves phosphodiester bond 5′ to the lesion by hydrolytic mechanism. To determine the length of cleavage products we generated size markers by incubation of the 5′-[32P]-labelled 24 mer regular duplex DNA oligonucleotide with the MtbXthA protein, an AP endonuclease from M. tuberculosis, which contains robust 3′→5′ exonuclease activity (Fig. 1, lanes 4–6). As expected, incubation of 24 and 30 mer AA•T=T and AA•T-T duplexes with T4-PDG and UVDE resulted in the generation of 12 mer and 14 mer cleavage products, respectively (lanes 2, 3, 9 and 11–12), indicating the presence of UV adducts at positions 12 and 14 in 24 and 30 mer DNA duplexes, respectively. It should be noted that due to difference in the mechanism of actions between DNA glycosylase and endonuclease, the 12 and 14 mer cleavage products generated by T4-PDG migrated slower than the corresponding products generated by UVDE (lanes 2 vs. 3 and 11 vs. 12). T4-PDG excises damaged base and then cleaves the remaining AP site via β-elimination to generate the slowly migrating cleavage fragments containing the 3′-phospho-α,β-unsaturated aldehyde group (Dodson, Michaels & Lloyd, 1994). At the same time, UVDE generates faster migrating cleavage fragments with 3′-OH ends, which is expected from the hydrolytic mechanism of action of AP endonucleases (Hegde, Hazra & Mitra, 2008).

Figure 1 Analysis of the cleavage products generated by T4-PDG and UVDE enzymes.

T4-PDG and UVDE enzymes were acted upon 5′-[32P]-labelled 24 and 30 mer duplex oligonucleotides containing CPD and 6-4PP adducts. Lanes 1, 7 and 10, control non-treated oligonucleotides; lanes 2–3 and 8–9, 24 mer AA•T=T and AA•T-T duplexes incubated either with T4-PDG or UVDE; lanes 11 and 12, 30 mer AA•T=T duplex incubated with T4-PDG and UVDE, respectively; lanes 4–6, 24 mer regular AA•TT duplex incubated with MtbXth, a 3′-5′ exonuclease, to generate size markers. For details see materials and Methods.

E. coli MutY did not excise adenine opposite UV-induced adducts in oligonucleotide DNA duplexes

To examine whether aberrant BER could be involved in the processing of UV-induced DNA damage, the 24 mer AA•T=T and AA•T-T duplexes, in which non-damaged DNA strand containing A opposite the lesion is radioactively labelled with 5′-[32P], were incubated in the presence of E. coli MutY. After reactions the resulting AP sites were cleaved by light piperidine treatment and the products were separated on the denaturing PAGE. To determine the position of cleavage site in the non-damaged strand, we generated DNA size markers using 5′-[32P]-labelled 24 mer duplexes containing either single A•G mispair at position 12, referred as AA•GT, or single A•8oxoG mispair at position 11, referred as AA•ToG (where oG denotes 8oxoG). In addition, we constructed 24 mer single-stranded oligonucleotides with single Uracil (U) residue at positions 11 or 12, referred as 24UA and 24AU, respectively. As expected, incubation of 24 mer AA•ToG duplex with MutY and 24UA with E. coli UNG generated 10 mer cleavage fragment (Fig. 2, lanes 6, 8 and 11), whereas incubation of 24 mer AA•GT duplex and 24AU with the same enzymes generated 11 mer cleavage fragment (Fig. 2, lanes 2 and 12, and Fig. 3, lane 8). These results indicate that MutY excises mismatched A residues opposite 8oxoG and G. Importantly, no cleavage products were observed after incubation of AA•T=T and AA•T-T duplexes, indicating that MutY failed to recognize A residues opposite CPD and 6–4PP in 24 mer duplexes (Fig. 2, lanes 4 and 10, and Fig. 3, lane 10).

Figure 2 Analysis of the cleavage products generated by MutY and UNG when acting upon 24 mer oligonucleotides containing base modifications.

The 5′-[32P]-labelled 24 mer duplex oligonucleotides containing CPD, 6–4PP, mismatches A•G and A•8oxoG and single-stranded 24 mer oligonucleotide containing uracil were incubated with MutY and UNG, respectively. Lanes 1–10, 24 mer duplexes incubated or not with MutY; lanes 11–14, 24 mer single-stranded oligonucleotides containing single Uracil residue, incubated or not with Ung to generate size markers. For details see Materials and Methods.

Figure 3 Analysis of the cleavage products generated by MutY when acting upon 30 and 24 mer duplex oligonucleotides containing either A•G mismatch or CPD adduct.

MutY was acted upon 5′-[32P]-labelled 30 and 24 mer duplex oligonucleotides containing either A•G mismatch or CPD adduct. Lanes 1, 3 and 5, control non-treated 30 mer oligonucleotides; lanes 2, 4 and 6, 30 mer duplexes incubated with MutY; Lanes 7, 9 and 11, control non-treated 24 mer duplexes; lanes 8, 10 and 12, 24 mer duplexes incubated with MutY. For details see materials and Methods.

Next, we examined whether MutY could excise Adenines opposite CPD in an oligonucleotide duplex with different sequence context. For this, the 30 mer AA•T=T duplex, in which the complementary non-damaged DNA strand containing A opposite the lesion is radioactively labelled with 5′-[32P], was incubated in the presence of E. coli MutY. We generated DNA size markers using 5′-[32P]-labelled 30 mer duplexes containing either single A•G mispair at positions 15 and 16, referred as AA•GT and AA•TG, respectively. As expected, incubation of 30 mer AA•GT and AA•TG duplexes with MutY generated 15 and 14 mer cleavage fragments, respectively (Fig. 3, lanes 2 and 4). Again, no cleavage products were observed after incubation of AA•T=T duplex with MutY, indicating that MutY does not recognize A residues opposite CPD placed in two different sequence contexts (lane 6). In addition, we measured MutY activity on the mismatched 24 and 30 mer oligonucleotide duplexes containing CPD opposite four possible bipurine dinucleotides: GG, AG, GA and AA. The results showed no cleavage products after incubation of GG•T=T, AG•T=T, GA•T=T and AA•T=T duplexes with MutY (Fig. S1), indicating that the adenine-DNA glycosylase does not recognize A or G residues opposite CPD in the mismatched DNA duplexes.

Still, one cannot exclude that MutY may act on Adenines opposite other bipyrimidine photoproducts such as C=T, T=C and C=C. To examine this, we incubated heavily UV-irradiated covalently closed circular plasmid DNA (ccc) with MutY, APE1 and T4 PDG. Upon cleavage at the site of damage by a DNA repair enzyme, the ccc form is converted either to an open circular form or to a linear double-stranded fragment and these three forms can be separated and quantified by electrophoresis in agarose gel (Ishchenko et al., 2003). The results showed that while T4 PDG cleaves 97% of the UV-irradiated plasmid, MutY (both in the absence and in the presence of APE1) cleaves only tiny fraction of the plasmid (Fig. S2). These results further substantiate the fact that MutY has very little or no activity toward UV adducts and Adenines on non-damaged strand in UV-irradiated DNA.

UV-induced RifS→RifR mutations in MutY and NER-deficient E. coli mutants

Genetic and biochemical studies of E. coli have established that both the RecA-dependent recombination and NER pathway participate in the removal of UV-induced DNA damage. Subsequently, it was demonstrated that UV-DNA adducts can be bypassed by UmuD′2C-RecA-ATP complex in vitro with high efficiency (Jiang et al., 2009). Importantly, in the absence of functional NER system, the frequency of UV-induced mutagenesis for a given dose increases further because of the persistence of unrepaired UV adduct in DNA which in turn stimulate mutations via UmuDC-dependent pathway. To examine a potential role of the MutY-catalyzed DNA glycosylase activity in UV-induced mutagenesis, we assessed the sensitivity and mutagenesis of the E. coli MutY/MicA (micA) and NER (uvrA6) deficient strains exposed to 254 nm UV light. Previously, we hypothesized that the aberrant BER pathway induces mutation in replication-independent manner (Talhaoui et al., 2014). Furthermore, it has been demonstrated that UV-induced DNA lesions can undergo mutation fixation by the NER-induced mutagenesis (NERiM) in DNA replication-independent manner (Janel-Bintz et al., 2017). Therefore, to favor the detection of mutations occurring in a DNA replication-independent manner, we measured UV-induced mutagenesis in stationary phase cultures of the E. coli WT CC104 and AB1157 (WT), MutY-deficient BH980 (micA::KnR), NER-deficient AK146 (uvrA6) and double NER/MutY-deficient BH1220 (uvrA6 micA) strains. Data from a typical experiment are shown in Table 2, exposure of the CC104 WT strain to high doses of UV (180 J·m−2) resulted in 64-fold increase in the appearance of RifR colonies as compared to that of non-irradiated strain. At the same time, BH980 strain, isogenic to CC104, but micA exhibited only 11-fold increase. Similarly, exposure of AK146 strain to low dose of UV (10 J·m−2) resulted in 81-fold increase of RifS→RifR mutations frequencies as compared to the non-irradiated control (Table 2). Intriguingly, exposure of NER/MutY-deficient BH1220 strain to low dose of UV (10 J·m−2) showed only weak 3.9-fold increase in frequency of UV-induced RifR colonies as compared to the non-irradiated control. It should be stressed, that MutY-deficient strains (micA) exhibited 11- to 80-fold increase in spontaneous mutation rates as compared to MutY-proficient strains (Table 2). Combining data from several independent experiments in Fig. 4 demonstrated that WT and NER-deficient E. coli strains exhibited 45- to 220-fold increase in the UV-induced mutation frequencies, as compared to moderate 2.3- and 13-fold increase for MutY/NER and MutY-deficient strains, respectively. Taken together, these results suggest that in the absence of the MutY protein, bacterial cells exhibited low to moderate induction of the UV-induced RifS→RifR mutations frequencies.

Table 2 Frequencies of UV induced RifS→RifR mutation in E. coli WT versus micA strains.

E. coli strains	Dose J·m–2	Survival (%)	No UV	UV	No UV	UV	Fold increase of RifR mutants	
			LB 107 cells/mL	RifR 1 mL	LB 107 cells/mL	RifR 1 mL	RifR/LB (10−7)	RifR/LB (10−7)		
CC104 (WT)	180 J·m−2	5.1	3,600	210	200	7,670	0.058	3.73	64	
BH980 (micA::KanR)	180 J·m−2	12.0	200	129	87	640	0.645	7.37	11.4	
AB1157 (WT)	100 J·m−2	0.1	320	3,5	140	476	0,01	3,4	311	
AK146 (uvrA6)	10 J·m−2	5.2	255	5	19,5	31	0,0196	1,59	81	
BH1220 (uvrA6 micA::KanR)	10 J·m–2	1.0	300	246	35	112,5	0,82	3,21	3.9	

Figure 4 Graphic representation of the UV-induced increase in mutation frequencies in E. coli cells.

NER-proficient strains were exposed to dose 100–180 J·m−2 UV and NER deficient strains to doses 10 J·m−2 UV, only. Data from at least three experiments were used for statistical analysis.

Molecular spectra of spontaneous and UV-induced RifS→RifR mutations in E. coli WT and micA strains

To further examine whether MutY participates in the aberrant processing of the unrepaired UV lesions in vivo, we measured UV-induced mutagenesis in the NER/MutY-deficient E. coli BH1220 and BH1070 (uvrA6 micA) strains harbouring vector pJWT21-4 coding either for MutY-WT or for catalytically inactive mutant MutY-D138N. It should be noted that the BH1220 and BH1070 strains expressing MutY-WT exhibited very low rate of spontaneous mutations as compared to the same strains expressing mutant MutY-D138N. As expected, exposure of the BH1220 and BH1070 strains with MutY-WT to UV (10 J·m−2) resulted in more than 100-fold increase in RifS→RifR mutation frequencies (Fig. 5). In contrast, for the same dose of UV, these same strains but containing MutY-D138N mutant exhibited only three to sevenfold increase in the UV-induced mutations (Fig. 5). It is well established that the major type of mutations induced by UV light are C→T transitions. Since the majority of spontaneous mutations in MutY-deficient strains are due to oxidative damage to guanines in DNA, we suggested that the majority of UV-induced mutations in these strains would be due to cytosine-containing pyrimidine dimers in DNA and not due to guanine oxidation.

Figure 5 Graphic representation of the UV-induced increase in mutation frequencies in E. coli strains containing the WT and D138N mutant MutY protein.

NER-proficient AB1157 strain was exposed to 100 J·m−2 UV and NER deficient strains to only 10 J·m−2 UV. Data from at least three experiments were used for statistical analysis.

To examine changes in the mutation spectra in WT and DNA repair-deficient E. coli strains after UV exposure, we performed DNA sequencing of RpoB gene from RifR clones obtained before and after UV exposure of the cells. As expected, analysis of the spontaneous and UV-induced mutation spectra in the E. coli WT, uvrA6 and uvrA6 micA strains revealed that all strains exhibit dramatic increase in the frequency of C•G→T•A transition: from 12.6% to 81% in WT and from 2.6–6.5% to 64–76% in uvrA6 micA strains (Table 3). It should be noted that BH1070 and BH1220 strains, as compared to the MutY-proficient strains, exhibited moderate increase in the overall mutation frequencies after UV exposure, possibly because of the masking effect of high spontaneous mutation rate in non-exposed cells which is primarily due to G•C→T•A transversions (which make >80% of all mutations). Remarkably, UV irradiation dramatically changes the mutation spectra in these two uvrA6 micA strains with majority of mutations being C•G→T•A transitions (which make 64–76% of all UV-induced mutations after UV exposure (Table 3). Using these mutation spectra, we calculated the relative increase in the frequency of C•G→T•A transitions after UV irradiation. The results revealed that NER/MutY-deficient BH1220 and BH1070 strains exhibited dramatic 86- and 239-fold increase, in the frequency of C•G→T•A transitions after UV exposure, although the overall increase in RifS→RifR mutation frequencies were only 3.9- to 4.5-fold (Table 3).

Table 3 Frequencies and molecular spectra of spontaneous and UV induced mutations in the E. coli MutY-proficient and MutY-deficient strains.

E. coli strains	Base substitution	Mutation spectra in control cells (%)	Mutation spectra in % in UV irradiated cells (%)	Fold increase of C•G→T•A transitions after UV irradiation	Fold increase of RifR mutants after UV irradiation	
AB1157 (WT)	C→T & G→A	12.6	81.3	768	140	
T→C & A→G	38.4	3.1		
A→C & T→G	30.8	–		
A→T & T→A	7.6	12.5		
G→T & C→A	3.8	3.1		
BH200 (uvrA::Tn10)	C→T & G→A	40.0	92.0	224	98	
T→C & A→G	20	–		
A→T & T→A	20	–		
G→T & C→A	–	3.8		
AK146 (uvrA6)	C→T & G→A	60.0	75.0	218	224	
T→C & A→G	15.0	–		
G→T & C→A	15.0	12.5		
A→T & T→A	10.0	12.5		
BH1220 (uvrA6 micA::KanR)	C→T & G→A	6.5	63.6	86	3.9	
G→T & C→A	84.0	27.3		
BH1070 (uvrA6 micA::KanR)	C→T & G→A	2.6	76.2	239	4.5	
G→T & C→A	87.2	7.1		

Of note, the mutational signatures in the control non-irradiated strains varied depending on the genetic background. For example, of the most frequent base substitution in non-irradiated AB1157 (WT) was T•A→C•G transition followed by A•T→C•G transversion (Table 3). At the same time, C•G→A•T transitions were prevalent in the non-irradiated BH200 (uvrA::Tn10) and AK146 (uvrA6) strains.

Structural features counteracting the aberrant DNA substrate specificity of MutY

The reported opposite-base specificity of MutY includes naturally occurring bases 8oxoG, G, C (Radicella, Clark & Fox, 1988; Tsai-Wu, Liu & Lu, 1992), 8-oxoadenine (Bulychev et al., 1996) as well as artificial base analogs 8-oxohypoxanthine, 8-oxonebularine, 8-methoxyguanine, 8-thioguanine, 7-methyl-8-oxoguanine and 8-bromoguanine (Bulychev et al., 1996;Manlove et al., 2017). The structure of MutY from Geobacillus stearothermophilus (Bst-MutY) bound to DNA containing an A:8oxoG mispair reveals intrahelical 8oxoG forming specific hydrogen bonds with a conserved Ser residue, O8[8oxoG]…N[S308] and N7[8oxoG]…Oγ[S308] (Fromme et al., 2004; Lee & Verdine, 2009). However, this arrangement of hydrogen bonds is not possible with G and many other bases opposite the excised adenine. It has been suggested that for DNA glycosylases excising canonical bases from non-canonical base pairs, such as MutY, the decisive factor is selective destabilization of a non-canonical base pair and stabilization of the everted natural base in the enzyme-substrate complex (Talhaoui et al., 2017).

In order to rationalize the lack of activity of MutY on A opposite CPD, we have analyzed the structures of all available MutY-DNA complexes (Fromme et al., 2004; Lee & Verdine, 2009; Wang, Chakravarthy & Verdine, 2017; Wang, Lee & Verdine, 2015) and all DNA molecules containing a CPD (Biertumpfel et al., 2010; Fischer et al., 2011; Horikoshi et al., 2016; Li et al., 2004; McAteer et al., 1998; Park et al., 2002; Silverstein et al., 2010; Vasquez-Del Carpio et al., 2011; Vassylyev et al., 1995). By covalently linking two adjacent residues, a CPD places strong constraints on the geometry of the damaged nucleotides, keeping them closer than in regular B-DNA (C1′–C1′ distance 3.85 ± 0.29 Å in a CPD vs. 5.17 ± 0.45 Å in B-DNA). On the other hand, intrusion of an aromatic residue (Tyr88 in Bst-MutY) 5′ of 8oxoG stretches DNA in this region (C1′–C1′ distance 7.22 ± 0.61 Å) and slightly compresses it 3′ of 8oxoG (4.50 ± 0.23 Å). Thus, a covalently bound CPD is incompatible with the geometry requirements to the non-cleaved strand in a MutY-DNA complex targeting either 5′- or 3′-adenine in the AA dinucleotide opposite a CPD (Fig. 6).

Figure 6 Distance between C1′ atoms in the adjacent nucleotides.

(A) Graphical representation of the interatomic distances. In 1TTD, 1COC and 355D structures, all distances (except within a CPD in 1TTD) were measured as representative of B-DNA. MutY 5′ and MutY 3′, the distances from C1′ of oxoG to C1′ of 5′- and 3′-adjacent nucleotides, respectively (structures 1RRQ, 1RRS, 1VRL, 3FSP, 5DPK, 3G0Q, 4YOQ, 4YPH and 4YPR). CPD, the distance between two C1′ atoms within a CPD (structures 1N4E, 1SKS, 1SL1, 1SL2, 1TTD, 1VAS, 3MFI, 3MR3, 3MR5, 3MR6, 3PZP, 3SI8, 4A0A, 4A0B, 4A08, 4A09 and 5B24). (B) Close view of a CPD in two representative CPD-containing structures (1TTD: free CPD-containing DNA, 1VAS: CPD-containing DNA from a complex with phage T4 endonuclease V) and a structure of the non-target strand from a complex with G. stearothermophilus MutY (1RRQ). In the 1RRQ structure, the wedging Tyr88 residue is shown. C1′-C1′ distances are indicated.

Discussion

Recently, we showed that human TDG can target non-damaged DNA strand to remove mismatched T opposite deaminated/oxidized adenine residues in duplex DNA in TpG/CpA* context (where A* is a damaged adenine residue). This aberrant excision of a normal base initiates repair synthesis that uses damaged DNA template leading to T→C mutation fixation in the absence of DNA replication. This finding points to a possible role of other mismatch-specific DNA glycosylases which do not discriminate damaged versus non-damaged DNA strand when excising regular DNA bases such as bacterial MutY and human MUTYH proteins. Indeed, it was shown that E. coli MutY participates in mutagenic post-replicative excision of regular A opposite a misincorporated 8oxoG residue which results in the increased A•T→C•G transversion rates in both E. coli WT and mutT strains(Fowler et al., 2003). It has been also demonstrated that MUTYH is involved in aberrant processing of UV lesions and interferes with NER machinery (Mazouzi et al., 2017). Based on these observations we speculated that MutY/MUTYH similar to TDG/MBD4 may initiate aberrant excision of Adenines opposite damaged Thymine residues, for example CPD and 6–4PP adducts, which in turn may lead to futile repair and activation of the DNA damage response in the UV-exposed cells.

In the present study, we report that the E. coli MutY protein cannot excise A residues opposite UV-induced CPD and 6–4PP adducts in the duplex oligonucleotide under experimental conditions used. Importantly, we observed efficient MutY-catalyzed excision of mismatched A residues when opposite G and 8oxoG residues in duplex DNA with the same sequence contexts used for UV lesions. These biochemical results suggest that MutY and perhaps its human homologue MUTYH do not participate in UV-induced mutagenesis in bacterial and human cells, respectively. The inability of MutY to excise A from opposite a CPD seems to be due, at least in part, to a severe conformational restrictions inflicted by the covalent linkage between two pyrimidines: the distance between two thymines in a dimer is less than is required to fit into MutY active center and cannot be widened (Fig. 6). Thus, the pre-catalytic MutY-DNA complex with a CPD would be destabilized, and the efficiency of the reaction significantly compromised.

To clear out the possible role of MutY in the aberrant mutagenic repair, we examined whether UV-induced mutagenesis in E. coli depends on the presence of this mismatch-specific DNA glycosylase. We observed that after UV exposure the MutY-deficient E. coli strains exhibited lower induction of RifS→RifR mutations as compared to MutY-proficient strains (Table 2 and Fig. 4). It should be stressed that under normal conditions E. coli micA strains exhibit a 100-fold increase in the spontaneous mutation rates as compared to MutY/MicA-proficient strains (Nghiem et al., 1988). Actually, this spontaneous mutator phenotype of E. coli micA strains might conceal the mutagenic effect of UV light, which, under the experimental conditions used, induces mutations at similar rates. Thus, we have analyzed mutation spectra in E. coli strains before and after UV exposure to see whether the types of mutations changes in MutY-deficient cells. The DNA sequence analysis of RifR clones showed that the NER/MutY-deficient BH1220 and BH1070 strains exhibited dramatic 86- and 239-fold increase in the frequency of C•G→T•A transitions after UV exposure, respectively, although the overall increase in RifS→RifR mutation frequencies was only 3.9- and 4.5-fold, respectively (Table 3). These results indicate that the mutation spectra in MutY/MicA-proficient and -deficient cells change in similar manner after UV exposure. It appears that MutY does not play a significant role in the induction of C•G→T•A transitions after UV exposure in vivo. Nevertheless, recently, it has been demonstrated that the repair of UV-induced DNA lesions turns out to be mutagenic in non-dividing cells under certain circumstances. Indeed, Janel-Bintz et al., (2017) have demonstrated that in E. coli cells the removal of closely spaced UV lesions in the NER pathway can induce mutations which are not dependent on DNA replication. Interestingly, NERiM is functional in stationary phase cells and requires DNA polymerases IV and II. In line with these observations, it has been shown that active Pol V complex (UmuD’2C-RecA-ATP) formed after UV irradiation does not co-localize with replicative DNA polymerase III complexes (Robinson et al., 2015).

Although in the present study we did not observe aberrant repair activities of MutY in E. coli cells, the accumulated evidence shows that mammalian homologues of MutY are involved in the aberrant and futile DNA repair. Depletion of MUTYH by acetohexamide in XP cells promotes an alternative repair of UV-induced lesions and thereby increases cells survival (Mazouzi et al., 2017). Under enhanced oxidative stress and in the absence of OGG1, the MUTYH-mediated BER mechanism might become engaged in futile and cytotoxic repair steps (Nakabeppu, 2014; Sheng et al., 2012). Also, it has been demonstrated that MUTYH induces the persistent accumulation of SSBs in the nascent DNA strand which in turn promotes MLH1/PARP1-dependent human cell death (Oka et al., 2014), retinal inflammation and degeneration in a mouse model of Retinis pigmentosa (Nakatake et al., 2016). A possible role of the MUTYH-initiated aberrant BER in DNA damage response was suggested by the study of the mechanisms of neurodegeneration caused by 3-nitropropionic acid (3NP) (Sheng et al., 2012). The authors showed that 3NP induced oxidative stress results in the accumulation of 8oxoG and SSBs in mitonchondrial DNA of neurons. SSBs accumulation and neurodegeneration were alleviated in mutant mice lacking MUTYH. However, OGG1 and MTH1, an 8oxo-dGTP hydrolase, offered protection, suggesting that aberrant repair of the misincorporated adenine opposite 8oxoG in DNA by MUTYH leads to SSB accumulation, which in turn triggers mitochondrial impairment and retrograde signalling to the nucleus of neurons.

Conclusion

In the present study we further characterized the biochemical and genetic properties of the well-known bacterial MutY DNA glycosylase. The data demonstrate that bacterial MutY does not recognize A opposite damaged T residues such as pyrimidine dimers in duplex DNA and is not involved in UV-induced mutagenesis in E. coli. Based on these observations, we propose that the role of MUTYH, a human homologue of MutY, in the increased genotoxicity of UV damage in XP cells might not be due to its aberrant activity toward A residues in the non-damaged DNA strand. Although we did not find clear evidence for the involvement of MutY-like DNA glycosylases in the aberrant mutagenic repair, we cannot exclude that the non-dividing cells such as reversibly growth-arrested dormant hematopoietic stem cells or terminally differentiated neurons may be prone to the aberrant excision of regular DNA bases in damaged DNA duplexes by the mono-functional mismatch-specific DNA glycosylases. Additional studies are required to investigate whether MutY and MUTYH act on different DNA substrates with damaged Thymine residues.

Supplemental Information

Supplemental Information 1 Raw image of Figure 1. Analysis of the cleavage products generated by T4-PDG and UVDE enzymes when acting upon 5′-[32P]-labelled 24 and 30 mer duplex oligonucleotides containing CPD and 6-4PP adducts.

Lanes 1, 7 and 10, control non-treated oligonucleotides; lanes 2–3 and 8–9, 24 mer AA•T=T and AA•T-T duplexes incubated either with T4-PDG or UVDE; lanes 11 and 12, 30 mer AA • T=T duplex incubated with T4-PDG and UVDE, respectively; lanes 4–6, 24 mer regular AA•TT duplex incubated with MtbXth, a 3′-5′ exonuclease, to generate size markers. For details see Fig. 1 in the manuscript.

Click here for additional data file.

Supplemental Information 2 Raw image of Figure 2. Analysis of the cleavage products generated by MutY and Ung when acting upon 5′-[32P]-labelled 24 mer duplex oligonucleotides containing CPD, 6-4PP, Uracil and mismatches A•G and A• 8oxoG.

Lanes 1–10, 24 mer duplexes incubated or not with MutY; lanes 11–14, 24 mer single-stranded oligonucleotides containing single Uracil residue, incubated or not with Ung to generate size markers. For details see Fig. 2 in the manuscript.

Click here for additional data file.

Supplemental Information 3 Raw image of Figure 3. Analysis of the cleavage products generated by MutY when acting upon 5′-[32P]-labelled 30 and 24 mer duplex oligonucleotides containing either A•G mismatch or CPD adduct.

Lanes 1, 3 and 5, control non-treated 30 mer oligonucleotides; lanes 2, 4 and 6, 30 mer duplexes incubated with MutY; Lanes 7, 9 and 11, control non-treated 24 mer duplexes; lanes 8, 10 and 12, 24 mer duplexes incubated with MutY. For details see Fig. 3 in the manuscript.

Click here for additional data file.

Supplemental Information 4 Raw data for Figure 4. Graphic representationof the UV-induced increase in mutation frequencies in E. coli cells.

The values listed represent the fold increases in occurrence of RifR mutants after UV exposure. Statistical Analysis of data used in Fig. 4: Mean & Standard Deviation.

Click here for additional data file.

Supplemental Information 5 Raw data for Figure 5. Graphic representation of the UV-induced increase in mutation frequencies in E. coli strains containing the WT and D138N mutant MutY protein.

The values listed represent the fold increases in occurrence of RifR mutants after UV exposure. Statistical Analysis of data used in Fig. 5: Mean & Standard Deviation.

Click here for additional data file.

Supplemental Information 6 Raw data for Figure 6. Distance between C1’ atoms in the adjacent nucleotides.

The measurements were done on the coordinates in the indicated .pdb files, which are freely accessible at rcsb.org.

Click here for additional data file.

Supplemental Information 7 Supplementary Figure S1. Analysis of the cleavageproducts generated by MutY when acting upon 5′-[32P]-labelled 24 and 30 mer duplex oligonucleotides containing the G• T mismatch and CPD adduct.

Lanes 1, 3, 5, 7, 9, 11, 13, 15 and 17, control non-treated 24 and 30 mer duplex oligonucleotides; lanes 2, 4, 6, 8, 10, 12, 14, 16 and 18, 24 and 30 mer duplex oligonucleotides incubated with MutY. For details see materials and Methods.

Click here for additional data file.

Supplemental Information 8 Raw image of Supplementary Figure S1.

Analysis of the cleavage products generated by MutY when acting upon 5′-[32 P]-labelled 24 and 30 mer duplex oligonucleotides containing the G• T mismatch and CPD adduct. For details see materials and Methods see Fig. S1.

Click here for additional data file.

Supplemental Information 9 Supplementary Figure S2. Cleavage of the UV-irradiated pBlueScript SK(+) plasmid DNA by DNA repair enzymes.

(A) Agarose gel electrophoresis (0.8%) of the cleavage products generated by MutY, APE1 and T4 PDG when acting upon supercoiled (ccc) form of plasmid DNA. Lane 1, GeneRuler 1 kb DNA ladder; lanes 2–6, control non-treated plasmid DNA; lanes 7–11, UV-irradiated plasmid DNA. The arrows denote the position of “ccc”, “oc” and “lds” forms of plasmid DNA . For details see Materials and Methods. (B) Graphical representation of data from panel A.

Click here for additional data file.

Supplemental Information 10 Raw image of Supplementary Figure S2, panel A.

Cleavage of the UV-irradiated pBlueScript SK(+) plasmid DNA by DNA repair enzymes. (A) Agarose gel electrophoresis (0.8%) of the cleavage products generated by MutY, APE1 and T4 PDG when acting upon supercoiled (ccc) form of plasmid DNA. For details see Fig. S2.

Click here for additional data file.

Supplemental Information 11 Raw data for supplementary Figure S2, panel B.

Click here for additional data file.

We are grateful to S. Boiteux and J. H. Miller for the E. coli strains.

Additional Information and Declarations

Competing Interests

Author Contributions

Data Availability

The authors declare that they have no competing interests.

Caroline Zutterling performed the experiments, analyzed the data, prepared figures and/or tables, authored or reviewed drafts of the paper.

Aibek Mursalimov performed the experiments, analyzed the data, prepared figures and/or tables, authored or reviewed drafts of the paper.

Ibtissam Talhaoui performed the experiments, analyzed the data, prepared figures and/or tables, authored or reviewed drafts of the paper, dNA sequencing analysis to identify mutations.

Zhanat Koshenov performed the experiments, analyzed the data.

Zhiger Akishev performed the experiments, analyzed the data.

Amangeldy K. Bissenbaev conceived and designed the experiments, analyzed the data, contributed reagents/materials/analysis tools, authored or reviewed drafts of the paper.

Gerard Mazon conceived and designed the experiments, analyzed the data, contributed reagents/materials/analysis tools.

Nicolas E. Geacintov conceived and designed the experiments, contributed reagents/materials/analysis tools, oligonucleotides containing UV lesions.

Didier Gasparutto conceived and designed the experiments, contributed reagents/materials/analysis tools, oligonucleotides containing base modifications.

Regina Groisman conceived and designed the experiments, analyzed the data.

Dmitry O. Zharkov conceived and designed the experiments, performed the experiments, analyzed the data, contributed reagents/materials/analysis tools, prepared figures and/or tables, authored or reviewed drafts of the paper.

Bakhyt T. Matkarimov conceived and designed the experiments, analyzed the data, contributed reagents/materials/analysis tools, prepared figures and/or tables, authored or reviewed drafts of the paper.

Murat Saparbaev conceived and designed the experiments, analyzed the data, prepared figures and/or tables, authored or reviewed drafts of the paper, approved the final draft.

The following information was supplied regarding data availability:

The raw data is available in the Supplemental Information.

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
