# Peer review of "Aberrant repair initiated by the adenine-DNA glycosylase does not play a role in UV-induced mutagenesis in Escherichia coli"

_PeerJ, doi:10.7717/peerj.6029_

## Round 0.1 · original submission · Minor Revisions

Please revise the paper according to the reviewers' remarks. The paper will also need English editing.

Reviewer 1 ·

Basic reporting

This a detailed work on enzymology, in the usual style of the laboratory. The interest of the research is clearly stated in the introduction and the experimental design is ok for me. The combination of biochemical experiments with complementary analyses in vivo using E. coly mutants make the results solid.

Experimental design

I think the design is appropriate.
I only have a minor question for the authors. It has been reported that some endonucleolytic repair activities are quite dependent on the sequence context. In particular I recall a work by the authors in which they show that uracil may be a substrate for APE1-mediated NIR, but this strongly depends on the sequence context. I wonder whether the authors have check MutY activity in different sequence contexts, in particular, mismatched DNA duplexes (AG/T=T or similar).

Validity of the findings

Data is enough robust.

Additional comments

My only main concern is the title of the work. I think that the title may be misleading. Negative results are still interesting, but the title suggests that MutY indeed has a role in UV-induced mutagenesis, while the clear conclusion of the paper is that MutY does not participate in that process. I suggest a title in line with the final conclusion.
I also would like that more clear information would be included in the figures. For instance, in figure 1, it would be helpful to denote somehow the difference between the two substrates with T=T and the sequence context. Similarly, the authors should indicate that in figure 2 some lines are controls with ssDNA or the genotype of strains in figure 4.

Reviewer 2 ·

Basic reporting

The manuscript is generally well written, using clear English throughout, although there are a few minor grammatical errors in places which need correcting.

Experimental design

No comment.

Validity of the findings

No comment.

Additional comments

This manuscript describes aims to understand whether MutY plays a role in the repair of adenine residues opposite cyclobutane pyrimidine dimers (CPDs) and 6-4-photoproducts (6-4PP) generated by UV, and secondly whether MutY contributes to changes in mutation frequencies after UV irradiation. To my knowledge, this has not been investigated previously. The authors used a combination of in vitro biochemical assays using oligonucleotide substrates containing site specific CPD and 6-4PP damage in combination with MutY protein, and mutation frequency analysis in E.coli strains proficient or deficient in MutY following UV radiation. These experiments and results are all described and presented in sufficient detail.
The results from the study are largely against the hypothesis, but what these clearly demonstrate is that MutY is not capable of removing adenine opposite CPDs or 6-4PP in vitro from oligonucleotide substrates, but as expected is active in removal of adenine versus guanine or 8-oxoguanine (Figure 2 and 3). The authors also discovered that E.coli deficient in MutY do not harbour increased mutation frequencies in response to UV (Figure 5). However what they did find was an unexpected and interesting decrease in mutation frequency in MutY-deficient (BH980) and in MutY/NER (BH1220)-deficient strains. This decrease in mutation rates appeared to be dependent on the active protein, as demonstrated by complementation with wild type and not inactive MutY (Figure 5). The reason for this was not particular clear in the Discussion. Therefore my major comment would be to speculate on this finding given the observation that MutY clearly does not have an impact on repair of CPD or 6-4PP associated lesions at least in vitro.

Minor comments:
1. Some minor errors in grammar throughout the manuscript. For example, line 45 “regular bases resulted from“, should be “regular bases resulting from”; line 465 “evidences are accumulated”, should be “evidence has accumulated”. Line 486 should also state that “MutY does not recognise A opposite CPD or 6-4PP in duplex DNA”, to be more precise.
2. A correction to Figure 5 is required (is the MutY-D138N strain mislabelled as BH10701070 rather than BH1070?).
3. I suggest the incorporation of doses of UV radiation in Figures 4 and 5 for clarity, as these differ for the different strains.

Reviewer 3 ·

Basic reporting

No comments

Experimental design

No comments

Validity of the findings

No comments

Additional comments

While DNA repair pathways are guardians of the genome, aberrant activity of these pathways may increase mutagenesis. In the past, Saparbaev and colleagues have already characterized one such aberrant activity of a base excision repair pathway. In this work, considering the increased UV-sensitivity of human XP cells producing a MutY homolog, they tested both in vivo and in vitro the possibility that E. coli MutY will act aberrantly on the UV-lesions, cyclobutane pyrimidine dimers and 6-4 photoproducts.

They assembled an oligonucleotide-based experimental system to detect MutY action on the strand opposite the pyrimidine dimers, using A-G mismatches as positive controls, but detected no cleavage, in two different sequence contexts.

At the same time, they show that in vivo, the mutY defect significantly (~10-fold) reduces increase in mutagenesis after UV, in both NER-proficient and -deficient backgrounds. The authors point out that this effect is due to the ~10-fold increase in the spontaneous mutagenesis in the mutY mutants.
They explore this point further, by predicting and then demonstrating that before UV, ~85% of all base substitutions in the mutY mutants were the expected G—>T transversions, whereas after UV, ~70% of all base substitutions became C—>T transitions, indicative of TLS polymerase errors at pyrimidine dimers.

Finally, they consider published crystal structures and argue using measurements of inter-atomic distances that the structure of B-DNA around PD is significantly different from the structure of B-DNA around the A-oG mispair, apparently precluding recognition of the former by MutY enzyme.



Major points

(May be in the future, but…) I would also look at the opposite (PD-containing) strand for any MutY cleavage, just in case.


Minor points

It would be nice if the MS had page numbers. In the absence of these, I have assigned page 6 of the MS file (the title page) number 1, while all other pages are listed in relation to page 6. For example, Abstract is on page 3, while Introduction begins on page 5.

On page 13 (say, line 328), the authors should call mutY mutants this very name, or, at the least, should explain that micA is the mutY equivalent, — otherwise confusing. Also, the micA mutants are not BER-deficient, — they are deficient only in MutY!

On page 13 again, lines 340/341, I suggest to drop AB1157 results altogether, since the meaningful comparison here is between the uvrA and uvrA mutY mutants. This will allow to shorten the corresponding sentence and to combine the two current sentences on lines 340-345 into a single logical construct.

Page 18, lines 461-463 — this statement needs a citation (probably PMID: 26317348).

Fig. 6 — an explanatory panel showing the relevant structures would be helpful here.

Table 3 — in AB1157, the second category of base substitutions is T—>C, whereas for all other strains it is G—>T. Is this OK? It is not mentioned or rationalized anywhere…


Miscellaneous

Page 3, line 43 — insert comma after "exogenous factors"

Page 3, line 44 — insert comma after "do exist"

Page 3, line 45 — "resulting". Also, replace "and" with "or".

Page 3, line 50 — remove "which is". Also, replace "in" with "via"

Page 3, line 51 — Insert "However," before "MutY"

Page 3, line 60 — replace "cell survival" with "production of DNA repair enzymes"

Page 3, line 64 — replace "While" with "At the same time"

Page 5, line76 — insert two commas, one before "such as methylation", the other after

Page 5, line 80 — replace "in" with "via"

Page 5, line 82 — "presents"

Page 5, line 87 — replace "endogenous source of DNA damage" with "oxidative DNA lesion"

Page 5, line 89 — replace "damage" with "lesions"

Page 5, line 94 — replace "opposite to" with "across"

Page 5, line 98 — delete "were"

Page 6, line 104 — "Multiple evidence demonstrates"

Page 6, line 108 — Replace "Whereas" with "At the same time"

Page 6, line 110 — delete "dramatically"

Page 6, line 112 — replace "regular" with "correct". Also, delete "non-damaged". Also, replace "it's" with "it is".

Page 6, line 128 — delete "to"

Page 6, line 132 — replace "damage" with "lesions"

Page 7, line 140 — remove the comma and insert "the" instead

Page 7, line 145 — "evidence suggests"

Page 7, line 149 — replace "high-fidelity" with "replicative"

Page 7, line 154 — delete "to". Also, line 155. In general, delete "to" from combination "opposite to" throughout the text. I am stopping correcting this.

Page 7, line 156 — change "proposed" to "suspected"

Page 7, line 157 — replace "can" with "could"

Page 9, line 202 — replace "and" with "or"

Page 9, line 216 — replace "corresponding" with "the"

Page 10, line 232 — define PBS buffer

Page 11, line 261 — replace "for" with "in", in two places

Page 11, line 279 — insert "the" after "than"

Page 11, line 280 — "cleaves the remaining"

Page 11, line 282 — replace "Whereas" with "At the same time"

Page 12, line 292 — insert "the" after "reactions"

Page 12, line 298 — replace "and" with "or"

Page 13, line 339 — replace "Whereas" with "At the same time"

Page 13, line 340 — replace "Whereas" with "Similarly"

Page 14, line 348 — say: "strains exhibited 45-220-fold increase…"

Page 14, line 350 — replace "BER/NER and BER" with "mutY"

Page 14, line 363 — replace "Whereas" with "In contrast"

Page 14, line 366 — delete the comma

Page 17, line 422 — insert "the" in front of "absence"

Page 18, line 450 — insert comma before "and" and another one after "irradiation"

Page 18, line 453 — delete the first comma

Page 18, line 454 — replace "are" with "was"

Page 18, line 465 — "evidence accumulates"

Page 18, line 467 — delete "the"

Page 18, line 468 — "in the absence"

Page 19, line 478 — replace "were being protective" with "offered protection"

Page 19, line 479 — insert comma after "accumulation"

Page 25, line 717 — "doses"

---

## Round 0.2 · accepted · Accept

Thank you for carrying out the modifications proposed by the reviewers.

#